# Explicit Box Detection Unifies End-to-End Multi-Person Pose Estimation

**Jie Yang[1,2]\*, Ailing Zeng[1]†, Shilong Liu[1], Feng Li[1], Ruimao Zhang[2]†, Lei Zhang[1]**

[1]International Digital Economy Academy (IDEA).

[2]Shenzhen Research Institute of Big Data, The Chinese University of Hong Kong, Shenzhen

{zengailing,liushilong,lifeng,leizhang}@idea.edu.cn

{jieyang5@link, zhangruimao@}cuhk.edu.cn

## Abstract

This paper presents a novel end-to-end framework with Explicit box Detection for multi-person Pose estimation, called ED-Pose, where it unifies the contextual learning between human-level (global) and keypoint-level (local) information. Different from previous one-stage methods, ED-Pose re-considers this task as two explicit box detection processes with a unified representation and regression supervision. First, we introduce a human detection decoder from encoded tokens to extract global features. It can provide a good initialization for the latter keypoint detection, making the training process converge fast. Second, to bring in contextual information near keypoints, we regard pose estimation as a keypoint box detection problem to learn both box positions and contents for each keypoint. A human-to-keypoint detection decoder adopts an interactive learning strategy between human and keypoint features to further enhance global and local feature aggregation. In general, ED-Pose is conceptually simple without post-processing and dense heatmap supervision. It demonstrates its effectiveness and efficiency compared with both two-stage and one-stage methods. Notably, explicit box detection boosts the pose estimation performance by 4.5 AP on COCO and 9.9 AP on CrowdPose. For the first time, as a fully end-to-end framework with a L1 regression loss, ED-Pose surpasses heatmap-based Top-down methods under the same backbone by 1.2 AP on COCO and achieves the state-of-the-art with 76.6 AP on CrowdPose without bells and whistles. Code is available at https://github.com/IDEA-Research/ED-Pose.

## 1 Introduction

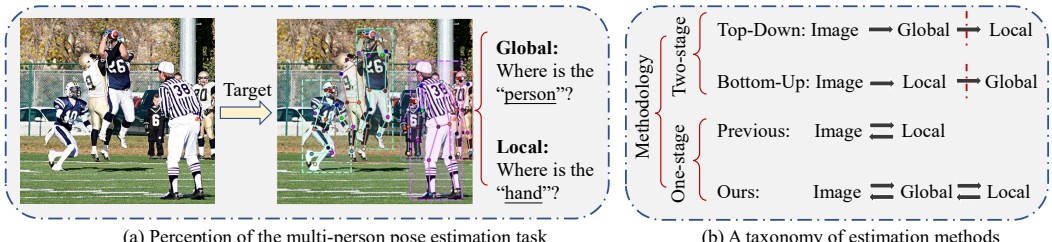

(a) Perception of the multi-person pose estimation task

(b) A taxonomy of estimation methods

Figure 1: Illustration of (a) the perception of the pose estimation task that usually captures global and local contexts concurrently; (b) a taxonomy of existing estimators. ED-Pose (Ours) is a novel one-stage method of learning both global and local relations in an end-to-end manner.

Multi-person human pose estimation has attracted much attention in the computer vision community for decades for its wide applications in areas of augmented reality (AR), virtual reality (VR), and human-computer interaction (HCI). Given an image, it targets to localize the 2D keypoint positions for every person in the image. Although many methods have been developed (Xiao et al., 2018;

---

\*This work was done when Jie Yang was intern at IDEA.

†Corresponding author.

Sun et al., 2019; Cheng et al., 2020; Mao et al., 2022; Shi et al., 2022), it remains challenging and intractable for situations with heavy occlusions, hard poses, and diverse body part scales.

Intuitively, as shown in Figure 1, this task needs to focus on both global (human-level) and local (keypoint-level) dependencies, which concentrate on different levels of semantic granularity. Mainstream solutions are normally two-stage methods, which divide the problem into two separate subproblems (e.g., the global person detection and local keypoint regression). Such solutions include Top-Down (TD) methods (Xiao et al., 2018; Sun et al., 2019; Li et al., 2021b; Mao et al., 2022) which are of high performance yet with a high inference cost and Bottom-Up (BU) solutions (Cao et al., 2017; Newell et al., 2017; Cheng et al., 2020) which are fast in inference yet with relatively lower precision. However, all of these methods are non-differentiable between their global and local stages due to hand-crafted operations, like Non-Maximum Suppression (NMS), Region of Interest (RoI) cropping, and keypoint grouping post-processing. Lately, Poseur (Mao et al., 2022) tries to directly apply top-down methods to an end-to-end framework and finds that there will be a significant performance drop (about **8.7** AP on COCO), indicating the optimization conflicts between the learning of global and local relations.

Exploring a fully end-to-end trainable method to unify the two disassembled subproblems is attractive and important. Inspired by the success of recent end-to-end object detection methods, like DETR (Carion et al., 2020), there is a surge of related approaches that regard human pose estimation as a direct set prediction problem. They utilize a bipartite matching for one-to-one prediction with Transformers to avoid cumbersome post-processings (Li et al., 2021b; Mao et al., 2021a; 2022; Stoffl et al., 2021; Shi et al., 2022). Recently, PETR (Shi et al., 2022) proposes a fully end-to-end framework to predict instance-aware poses without any post-processings and shows a favorable potential. Nevertheless, it directly uses a pose decoder with randomly initialized pose queries to query local features from images. The only local dependency makes keypoint matching across persons ambiguous and thus leading to inferior performance, especially for occlusions, complex poses, and diverse human scales in crowded scenes. Moreover, either two-stage methods or DETR-based estimators suffer from slow training convergence and need more epochs (e.g., train a model above a week) to achieve high precision. Additionally, the convergence speed of DETR-based methods is even slower than bottom-up methods (Cheng et al., 2020). We address the details in Sec.3.

Based on the above observations, this work re-considers multi-person pose estimation as two Explicit box Detection processes named ED-Pose. We realize each box detection by using a decoder and cascade them to form an end-to-end framework, which makes the model fast in convergence, precise, and scalable. Specifically, to obtain global dependencies, the first process detects boxes for all persons via human box queries selected from the encoded image tokens. This simple step can provide a good initialization for the latter keypoint detection to accelerate the training convergence. Then, to capture local contextual relations and reduce ambiguities in the feature alignment, we regard the following pose estimation task as a keypoint box detection problem, where it learns both box positions and local contents for each keypoint. Such an approach can leverage contextual information near a keypoint by directly regressing the keypoint box position and learning local keypoint content queries without dense supervision (e.g., heatmap). To further enhance the global-local interactivity among external human-human, internal human-keypoint, and internal keypoint-keypoint, we design interactive learning between human detection and keypoint detection.

Following the two Explicit box Detection processes, we can unify the global and local feature learning using the consistent regression loss and the same box representation in an end-to-end framework. We summarize the related methods from the supervisions and representations. Compared with previous works, ED-Pose is more conceptually simple. Notably, we find that explicit global box detection will gain **4.5** AP on COCO and **9.9** AP on CrowdPose compared with a solution without such a scheme. In comparison to top-down methods, ED-Pose makes the human and keypoint detection share the same encoders to avoid redundant costs from human detection and further boost performance by **1.2** AP on COCO and **9.1** AP on CrowdPose under the same ResNet-50 backbone. Moreover, ED-Pose surpasses the previous end-to-end model PETR significantly by **2.8** AP on COCO and **5.0** AP on CrowdPose. In crowded scenes, ED-Pose achieves the state-of-the-art with **76.6** AP (by 4.2 AP improvement over the previous SOTA (Yuan et al., 2021)) without any bells and whistles (e.g., without multi-scale test and flipping). We hope this simple attempt at explicit box detection, simplification of losses, and no post-processing to unify the whole pipeline could bring in new perspectives to further one-stage framework designs.

Table 1: Comparisons of existing estimators from the losses and representations of the Human and Keypoints. Our proposed ED-Pose unifies both Human and Keypoint detection under the consistent L1 regression loss and the box representation, making the end-to-end training simple yet effective.

| Methodology | | Human Loss | Keypoint Loss | Human Representation | Keypoint Representation |
|---|---|---|---|---|---|
| Two-Stage Methods | Top-Down | Regression | Heatmap | $(x, y, h, w)$ | $(x, y)$ |
| | Bottom-Up | - | Heatmap | - | $(x, y)$ |
| One-Stage Methods | Previous | - | Regression + Heatmap | - | $(x, y)$ |
| | Ours | Regression | Regression | $(x, y, h, w)$ | $(x, y, h, w)$ |

## 2 RELATED WORK

**One-stage Multi-Person Pose Estimation:** With the development of anchor-free object detectors (Tian et al., 2019b; Huang et al., 2015), DirectPose (Tian et al., 2019a) directly predicts instance-aware keypoints for all persons from an image. The direct end-to-end framework provides a new perspective to avoid the above cumbersome issues met in two-stage methods. Generally speaking, these methods densely locate a set of pose candidates, which consist of joint positions from the same person. FCPose (Mao et al., 2021b) builds upon dynamic filters (Jia et al., 2016) in compact keypoint heads to boost both accuracy and speed. Meanwhile, Inspose (Shi et al., 2021) designs instance-aware dynamic networks to adaptively adjust part of the network parameters for each instance. Nevertheless, these one-stage methods still need NMS to remove duplicates in the post-processing stage. To further remove such hand-crafted components, PETR (Shi et al., 2022) views pose estimation as a hierarchical set prediction problem and proposes the first *fully* end-to-end pose estimation framework with the advent of DETR (Carion et al., 2020).

**Detection Transformers:** For the first time, DETR (Carion et al., 2020) performs object detection in an end-to-end manner by using a set-based global loss that forces directly unique predictions via bipartite matching and a Transformer encoder-decoder architecture. It simplifies object detection as a direct set prediction problem, dropping multiple hand-designed components and prior knowledge. Due to the effectiveness of DETR and its varieties (*e.g.*, Deformable DETR (Zhu et al., 2020)), their frameworks have been widely transferred in many complex tasks, such as Mask DINO (Li et al., 2022b) for segmentation and PETR (Shi et al., 2022) for pose estimation. Following the top-down methods, PRTR (Li et al., 2021b) and TFPose (Mao et al., 2021a) adopt Detection Transformers to estimate the cropped single-person images as a query-based regression task. To capture the underlying output distribution and further improve performance in the regression paradigm, Poseur (Mao et al., 2022) brings Residual Log-likelihood Estimation (RLE) (Li et al., 2021a) into the DETR-based top-down framework, achieving the state-of-the-art performance of regression-based methods. For the one-stage manner, POET (Stoffl et al., 2021) utilizes the property of DETR to directly regress the poses (instead of bounding boxes) of all person instances in an image. Recently, PETR (Shi et al., 2022) designs a *fully* end-to-end paradigm with hierarchical attention decoders to capture both relations between poses and kinematic joints.

All the aforementioned methods take advantage of Detection Transformers and densely regress a set of poses (only local relations). However, they ignore the importance of introducing explicit box detection in pose estimation to model both global and local dependencies well.

## 3 RETHINKING ONE-STAGE MULTI-PERSON POSE ESTIMATION

**The Necessities of One-stage Methods:** For a long time, the two-stage paradigm dominates the mainstream methods for multi-person pose estimators. It can be generally divided into top-down methods and bottom-up methods. Top-Down (TD) methods (Xiao et al., 2018; Sun et al., 2019; Mao et al., 2022) disentangle the task into detecting and cropping each person with an object detector (*e.g.*, Mask RCNN (He et al., 2017)) in an image from the global (person-level) dependency and then conduct the single-person pose estimation via another model. They focus on local (keypoint-level) relation modeling and improving the accuracy of single-person pose estimation. However, these methods still suffer from 1). heavy reliance on the performance of the human detector, 2). redundantly computational costs for additional human detection and RoI operations, and 3). the separate training for the human detector and the corresponding pose estimator. Instead, Bottom-Up (BU) methods (Cao et al., 2017; Newell et al., 2017; Cheng et al., 2020) first detect all the keypoints in an instance-agnostic fashion. Next, they employ a heuristical grouping algorithm to associate the detected keypoints that belong to the same person, which improves efficiency. Even

so, the complicated grouping scheme makes bottom-up methods hard to handle heavy occlusions and multiple-person scales, resulting in inferior performance. More importantly, both of them suffer from non-differentiable optimization between the global and local features, which is not perceptive. Intuitively, one-stage methods could alleviate the above issues since all modules can be optimized in an end-to-end manner, and balance effectiveness and efficiency. Interestingly, the recent DETR-based top-down method Poseur (Mao et al., 2022) tries to directly apply it to a one-stage framework and finds that it suffers from a significant performance drop. It might be the optimization conflicts between the global and local dependency learning using the shared encoder. Thus, how to design a one-stage framework effectively is still challenging and questionable.

**The Bottlenecks of Existing One-stage Methods:** In terms of existing DETR-based methods, most of them still adopt the top-down framework, and improve the second single-person pose estimation via regarding it as a sequence prediction problem (Mao et al., 2021b; Shi et al., 2021). PETR is the first work to make the whole pipeline end-to-end without any post-processing. However, existing methods still have some limitations. First, all of them only utilize local dependencies to regress keypoints. Directly regressing keypoints for each person via pose queries is semantically ambiguous as the bottom-up strategies to find all keypoints from raw images instead of from cropped images. Second, the pose or keypoint queries proposed in the above methods are randomly initialized without utilizing previously extracted features, making the training phase slow and ineffective. Third, the keypoint representation as a point lacks contextual information when it queries from the encoded features, leading to feature misalignment. Last, the interactions among global-to-global, global-to-local, and local-to-local are complex, especially in crowd scenes. The current models do not pay attention to handling these complicated relations. In this article, we try to tackle the above issues by using unified box representations and regression losses in a one-stage process.

# 4 METHODOLOGY

## 4.1 OVERVIEW

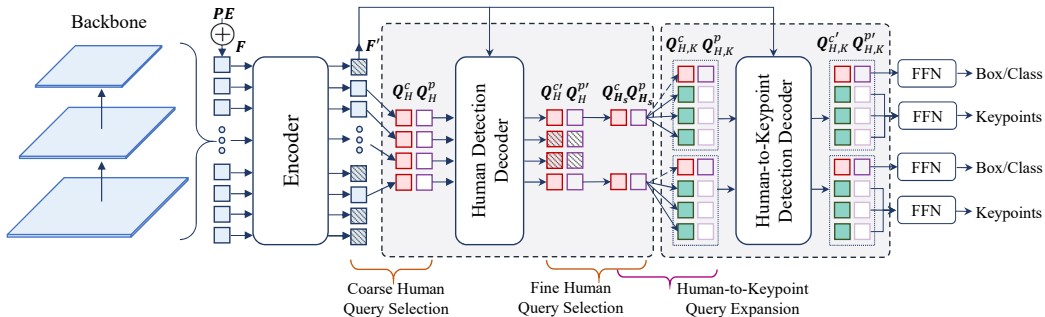

Figure 2: The overview architecture of our ED-Pose, which contains a Human Detection Decoder and a Human-to-Keypoint Detection Decoder to detect human and keypoint boxes explicitly.

As illustrated in Figure 2, the proposed ED-Pose is a fully end-to-end multi-person pose estimation framework without any post-processing. Given an input image, we first utilize a backbone to extract multi-scale features and obtain tokenized representations $\mathbf{F}$, and then feed them into the Transformer encoder (e.g., deformable attention modules (Zhu et al., 2020)) with Positional Embeddings ($\mathbf{PE}$) to calculate the refined tokens $\mathbf{F}'$. To improve the efficiency of the following decoding process, we conduct coarse human query selection from $\mathbf{F}'$ to obtain the sparse human content queries $\mathbf{Q}_H^c$. Then we utilize $\mathbf{Q}_H^c$ to generate the corresponding position queries $\mathbf{Q}_H^p$ via a Feed-Forward Network (FFN) and feed both $\mathbf{Q}_H^c$ and $\mathbf{Q}_H^p$ into the *Human Detection Decoder* to update into the corresponding queries $\mathbf{Q}_H^{c'}$ and $\mathbf{Q}_H^{p'}$, respectively (see Sec.4.2). For each output human content query, we attach a human box regression and class entropy supervision ($L_h^l$ and $L_c^l$) at the $l$-th decoder layer. Next, we further perform fine human query selection to discard the redundant human queries, obtaining content queries $\mathbf{Q}_{H_s}^c$ and position queries $\mathbf{Q}_{H_s}^p$. We initialize keypoint queries based on these retained high-quality human queries and concatenate the human and keypoint queries together to form human-keypoint queries. We name such a scheme as human-to-keypoint query expansion, and the obtained human-keypoint content queries $\mathbf{Q}_{H,K}^c$ and position queries $\mathbf{Q}_{H,K}^p$ are fed into *Human-to-Keypoint Detection Decoder*. In this way, we realize the keypoint detection at an instance level and update these queries to $\mathbf{Q}_{H,K}^{c'}$ and $\mathbf{Q}_{H,K}^{p'}$ layer by layer (see Sec.4.3). Also, the keypoint and human

box regression loss and class loss ($L_k^l$, $L_h^l$ and $L_c^l$) are added to the $l$-th decoder layer. Finally, we use several FFN heads to regress the keypoint positions in each human box.

**Loss:** Following the DETR (Carion et al., 2020), we employ a set-based Hungarian loss that forces a unique prediction for each ground-truth box and keypoint. Our overall loss functions contain classification $L_c$, human box regression $L_h$, and keypoint regression loss $L_k$. Notably, $L_k$ simply consists of the normal L1 loss and the constrained L1 loss named Object Keypoint Similarity (OKS) loss (Shi et al., 2022) without any dense supervision (e.g., heatmap).

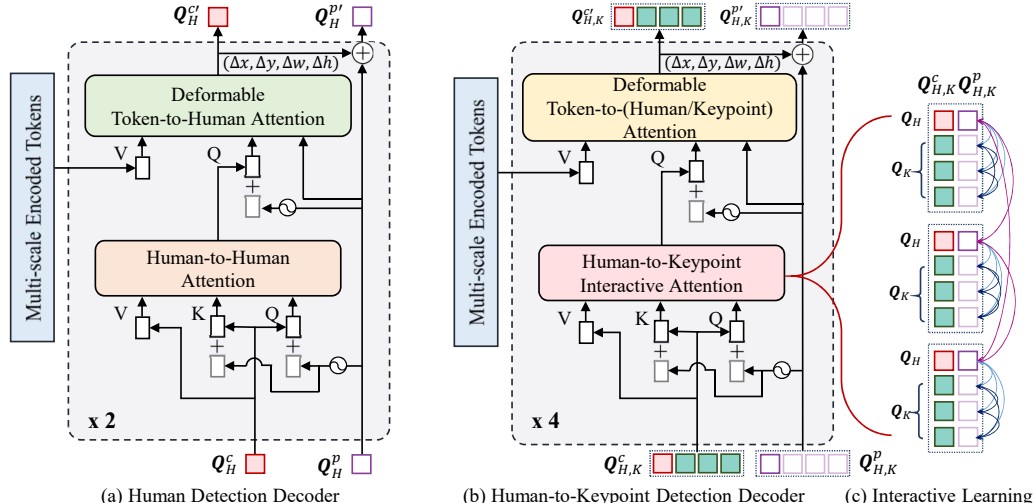

Figure 3: The detailed illustration of (a) Human Detection Decoder, (b) Human-to-Keypoint Detection Decoder and (c) the detailed Interactive Learning in Human-to-Keypoint Detection Decoder.

## 4.2   HUMAN DETECTION DECODER

In Figure 3 (a), the proposed Human Detection Decoder aims to predict the candidate bounding box positions for each person in the image, as well as the corresponding content representation by leveraging image context from all encoded tokens. Given $N$ input human content queries $\mathbf{Q}_H^c \in \mathbb{R}^{N \times D}$ and human position queries $\mathbf{Q}_H^p \in \mathbb{R}^{N \times 4}$, where $D$ is the channel dimension, the Human Detection Decoder outputs $N$ refined human box positions $\mathbf{Q}_H^{p'}$ and content representations $\mathbf{Q}_H^{c'}$.

**Coarse-to-Fine Human Query Selection**: Unlike previous work (Shi et al., 2022) to randomly initialize the input queries, we adopt the query selection (QS) strategy for better initialization (Zhang et al., 2022). Specifically, we propose a coarse-to-fine scheme to progressively select high-quality human queries. In practice, the *coarse human QS* selects first from a large number of refined tokens $\mathbf{F}' \in \mathbb{R}^{T \times D}$, where $T$ is the number of the tokens (e.g., about 15K). Thus, the human content queries $\mathbf{Q}_H^c \in \mathbb{R}^{N \times D}$ are initialized by the top-$N$ refined tokens' features, ranked by their instance classification scores. Here $N$ indicates the number of candidate human boxes (e.g., 900). Then we use these selected human content queries to calculate their position queries $\mathbf{Q}_H^p \in \mathbb{R}^{N \times 4}$ by employing a simple detection head. After passing through the Human Detection Decoder, we further conduct the *fine human QS* by only retaining $M$ (e.g., 100) refined human content queries and their position queries according to the classification scores, denoted as $\mathbf{Q}_{H_s}^c \in \mathbb{R}^{M \times D}$ and $\mathbf{Q}_{H_s}^p \in \mathbb{R}^{M \times 4}$.

**Human Box Detection**: As illustrated in Figure 3 (a), we combine $\mathbf{Q}_H^c$ with $\mathbf{Q}_H^p$ and feed them into Human-to-Human Attention (*i.e.* a self-attention layer) to calculate the relations among human queries, and output contextual enhanced ones. Motivated by the deformable DETR Zhu et al. (2020), these enhanced human content queries, together with their position queries are further fed into Deformable Token-to-Human Attention (*i.e.* a cross-attention layer) to update human queries by conducting interaction with multi-scale tokens. Next, to adjust the human box positions, we leverage updated human queries to calculate the 4D offsets following the DAB-DETR (Liu et al., 2022), and add them back to previous human position queries. In this way, the Human Detection Decoder refine the human content and position queries progressively by stacking multiple aforementioned layers and output $\mathbf{Q}_H^{c'} \in \mathbb{R}^{N \times D}$ and $\mathbf{Q}_H^{p'} \in \mathbb{R}^{N \times 4}$.

### 4.3 HUMAN-TO-KEYPOINT DETECTION DECODER

To unify both human and keypoint representations as the box and facilitate follow-up interactive learning, we regard multi-person pose estimation as the multiple set keypoint box detection problems. Based on the selected high quality human queries $\mathbf{Q}_{H_s}^p$ and $\mathbf{Q}_{H_s}^c$, we initialize multiple sets of keypoint queries (*i.e.* including both content and position queries) through the Human-to-Keypoint Query Expansion process, and concatenate the human and keypoint queries together as inputs of the Human-to-Keypoint Detection Decoder (Figure. 3 (b)) to calculate the precise keypoint positions for each person.

**Human-to-Keypoint Query Expansion**: After obtaining $\mathbf{Q}_{H_s}^p$ and $\mathbf{Q}_{H_s}^c$, the keypoint content query and positional query are initialized by the relevant human query instead of random initialization. In practice, we first initialize a set of learnable keypoint embeddings $\mathbf{V_e} \in \mathbb{R}^{1 \times K \times D}$ where $K$ is the total number of keypoints (e.g., 17). To concretize these embeddings to multiple specific persons' keypoint queries, we broadcast the first dimension of $\mathbf{V_e}$ to the number of human queries $M$ and then add it with $\mathbf{Q}_{H_s}^c \in \mathbb{R}^{M \times 1 \times D}$ to obtain $M$ set of keypoint content queries. For a specific set of keypoint position queries, we separate the initialization process into center coordinate initialization (e.g., {x, y}) and box size initialization (e.g., {w, h}). For the former, we adopt an FFN to regress all of the $K$ positions' coordinates by employing the corresponding human content query. For the latter, the sizes of $K$ boxes are conditioned by the width and height of the corresponding human box via dot-multiplying dynamic weights $\mathbf{W} \in \mathbb{R}^{K \times 2}$. After that, we consider each human box and the corresponding set of keypoint boxes as a whole and generate the human-keypoint content queries $\mathbf{Q}_{H,K}^c \in \mathbb{R}^{(M+M*K) \times D}$ and position queries $\mathbf{Q}_{H,K}^p \in \mathbb{R}^{(M+M*K) \times 4}$ for further predictions.

**The Interactive Learning between Human and Keypoint Detection**: As illustrated in Figure 3 (b) and (c), after generating human-keypoint queries, we feed them into Human-to-Keypoint Interactive Attention to learn relations from internal human-keypoint, internal keypoint-keypoint, and external human-human. Such an interactive learning method between global and local feature aggregations has been ingeniously introduced to ensure the effectiveness of keypoint feature extraction. In practice, we find that the fully connected interactive learning with external keypoint-keypoint would cause a large disturbance, especially in crowded situations, similar to problems encountered in bottom-up methods. In contrast, our external human-human interactions effectively propagate global context across different candidates and distinguish their relations clearly. Then we take the enhanced human-to-keypoint content queries to conduct the interaction with multi-scale encoded tokens to obtain updated human-to-keypoint queries. By stacking multiple layers illustrated in Figure 3 (b), we finally obtain the refined human-to-keypoint queries, denoted as $\mathbf{Q}_{H,K}^{c'}$ and $\mathbf{Q}_{H,K}^{p'}$.

## 5 EXPERIMENTS

Due to the page limit, we leave the detailed experiment setup in A, comparison results on COCO test-dev in B, comparisons on human detection in C, more qualitative results and analyses in D, and more discussion for ED-Pose (in Sec. E).

### 5.1 RESULTS ON CROWDPOSE

We first verify the effectiveness of ED-Pose with other state-of-the-art methods on the CrowdPose `test` set in Table 2. Compared with the top-down methods, we surpass the SimpleBaseline (Xiao et al., 2018) under the same backbone by 9.1 AP. we also show superiority on all previous methods combined with Swin-L (Liu et al., 2021) backbone and outperform PETR by 1.5 AP under the same backbone. When we enlarge the scales of backbones, ED-Pose will achieve the state-of-the-art 76.6 AP without multi-scale and flip tests.

### 5.2 RESULTS ON COCO

We further make comparisons with the state-of-the-art methods on COCO `val2017` and `test-dev` in Table. 3 and Table. 8 respectively. In general, our ED-Pose outperforms all existing bottom-up methods and one-stage methods under the same backbone without any tricks, even the dense heatmap-based top-down methods. The proposed method achieves 71.6 AP (by 2.8 AP improvement) with a $51.4\%$ inference time reduction via the ResNet-50 backbone. Additionally, our best model with the Swin-L* backbone achieves a 75.8 AP, showing a consistent performance improvement. The improvements in the `test-dev` are similar.

Table 2: Comparisons with state-of-the-art methods on CrowdPose `test` dataset. **TD, BU, OS** mean top-down, bottom-up, and one-stage methods, respectively. We use "HM." and "R." for heatmap-based losses and regression losses. † denotes the flipping test. The model with * is pre-trained on Objects365 (Shao et al., 2019a) with 5 feature scales. The underlined highlights the compared results. The **best results** are highlighted in **bold**.

| | Method | Loss | AP | $AP_{50}$ | $AP_{75}$ | $AP_E$ | $AP_M$ | $AP_H$ |
|---|---|---|---|---|---|---|---|---|
| **TD** | Sim.Base. (ResNet-50) | HM. | 60.8 | 81.4 | 65.7 | 71.4 | 61.2 | 51.2 |
| | HRNet (HRNet-w48) † | HM. | 71.3 | 91.1 | 77.5 | 80.5 | 71.4 | 62.5 |
| | TransPose-H | HM. | 71.8 | 91.5 | 77.8 | 79.5 | 72.9 | 62.2 |
| | HRFormer-B | HM. | 72.4 | 91.5 | 77.9 | 80.0 | 73.5 | 62.4 |
| **BU** | HrHRNet-w32† | HM. | 65.9 | 86.4 | 70.6 | 73.3 | 66.5 | 57.9 |
| | DEKR (HrHRNet-w32)† | HM. | 65.7 | 85.7 | 70.4 | 73.0 | 66.4 | 57.5 |
| | SWAHR (HrHRNet-w32)† | HM. | 71.6 | 88.5 | 77.6 | 78.9 | 72.4 | 63.0 |
| **OS** | PETR (Swin-L) | R.+HM. | 71.6 | 90.4 | 78.3 | 77.3 | 72.0 | 65.8 |
| | ED-Pose (ResNet-50) | R. | 69.9↑9.1 | 88.6 | 75.8 | 77.7 | 70.6 | 60.9 |
| | ED-Pose (Swin-L) | R. | 73.1↑1.5 | 90.5 | 79.8 | 80.5 | 73.8 | 63.8 |
| | ED-Pose (Swin-L*) | R. | **76.6**↑5.0 | **92.4** | **83.3** | **83.0** | **77.3** | **68.3** |

Table 3: Comparisons with state-of-the-art methods on COCO `val2017` dataset. † denotes the flipping test. ‡ removes the prediction uncertainty estimation in Poseur as a fair regression comparison. The underlined highlights the compared results. The inference time of all methods is tested on an A100, except that the detector of top-down methods is tested by the MMdetection (*i.e.*, 45ms).

| | | Method | Backbone | Loss | AP | $AP_{50}$ | $AP_{75}$ | $AP_M$ | $AP_L$ | Time [ms] |
|---|---|---|---|---|---|---|---|---|---|---|
| **Two-stage** | **TD** | Sim.Base.[†] | ResNet-50 | HM. | 70.4 | 88.6 | 78.3 | 67.1 | 77.2 | 45+86 |
| | | HRNet[†] | HRNet-w32 | HM. | 74.4 | 90.5 | 81.9 | 70.8 | 81.0 | 45+112 |
| | | PRTR[†] | ResNet-50 | R. | 68.2 | 88.2 | 75.2 | 63.2 | 76.2 | 45+85 |
| | | Poseur | ResNet-50 | RLE[‡] | 70.0 | - | - | - | - | 45+82 |
| | | Poseur | ResNet-50 | RLE | 74.2 | 89.8 | 81.3 | **71.1** | 80.1 | 45+82 |
| | **BU** | HrHRNet[†] | HRNet-w32 | HM. | 67.1 | 86.2 | 73.0 | 61.5 | 76.1 | 322 |
| | | DEKR[†] | HRNet-w32 | HM. | 68.0 | 86.7 | 74.5 | 62.1 | 77.7 | - |
| | | SWAHR[†] | HRNet-w32 | HM. | 68.9 | 87.8 | 74.9 | 63.0 | 77.4 | - |
| **E2E TD** | | Mask R-CNN | ResNet-101 | HM. | 66.0 | 86.9 | 71.5 | - | - | - |
| | | PRTR[†] | ResNet-101 | HM. | 64.8 | 85.1 | 70.2 | 60.4 | 73.8 | - |
| | | Poseur[†] | ResNet-101 | RLE | 68.6 | 87.5 | 74.8 | - | - | - |
| | | Poseur[†] | HRNet-w48 | RLE | 70.1 | 88.0 | 76.5 | - | - | - |
| **OS** | | InsPose | ResNet-50 | R.+HM. | 65.2 | 87.2 | 71.3 | 60.6 | 72.2 | 78 |
| | | PETR | ResNet-50 | R.+HM. | 68.8 | 87.5 | 76.3 | 62.7 | 77.7 | 105 |
| | | PETR | Swin-L | R.+HM. | 73.1 | 90.7 | 80.9 | 67.2 | 81.7 | 206 |
| | | ED-Pose | ResNet-50 | R. | 71.6↑2.8 | 89.6 | 78.1 | 65.9 | 79.8 | 51↓51.4% |
| | | ED-Pose | Swin-L | R. | 74.3↑1.2 | 91.5 | 81.6 | 68.6 | 82.6 | 88↓57.3% |
| | | ED-Pose | Swin-L* | R. | **75.8**↑2.7 | **92.3** | 82.9 | 70.4 | **83.5** | 142↓31.1% |

### 5.2.1 COMPARISON OF EFFECTIVENESS

**Comparison with one-stage methods**: Our method significantly outperforms all existing one-stage methods, especially in Table. 8, such as DirecetPose (Tian et al., 2019a), FCPose (Mao et al., 2021b), Inspose (Shi et al., 2021), CenterNet (Zhou et al., 2019), and PETR (Shi et al., 2022), showing that ED-Pose with explicit box detection is an effective solution for the end-to-end framework.

**Comparison with two-stage methods**: For bottom-up methods, ED-Pose outperforms state-of-the-art methods by a large margin, such as HigherHRNet (Cheng et al., 2020), DEKR (Geng et al., 2021), and SWAHR (Luo et al., 2021). Specifically, ED-Pose significantly surpasses the recently proposed SWAHR by **3.7** AP (71.6 AP *vs.* 67.9 AP) even with a much smaller backbone (ResNet-50 *vs.* HRNet-w32). For top-down methods, ED-Pose is 1.2 AP and 3.4 AP higher than SimpleBaseline (Xiao et al., 2018) (Sim.Base.) and PRTR (Li et al., 2021b), respectively. To the best of our knowledge, this is the first time that a fully end-to-end framework can surpass top-down methods.

**Comparison with end-to-end top-down methods**: Poseur (Mao et al., 2022) extends its framework to end-to-end human pose estimation with Mask-RCNN. Similarly, PRTR presents its end-to-end variant. However, compared to their two-stage paradigms, their end-to-end frameworks produce substantially inferior performance. Our fully end-to-end framework achieves much higher AP scores than theirs, indicating the effectiveness of our proposed methods.

**Qualitative results:** With explicit box detection, ED-Pose can perform well on a wide range of poses, containing viewpoint change, occlusion, motion blur, and crowded scenes. Fig. 4 demonstrates the results of the detected person and the corresponding keypoints. As can be observed, the human boxes are precise under many severe cases, thus they can provide effective global human information for further keypoint detection. The local regions of the keypoint boxes are also reasonable for bringing abundant contextual information near the keypoint.

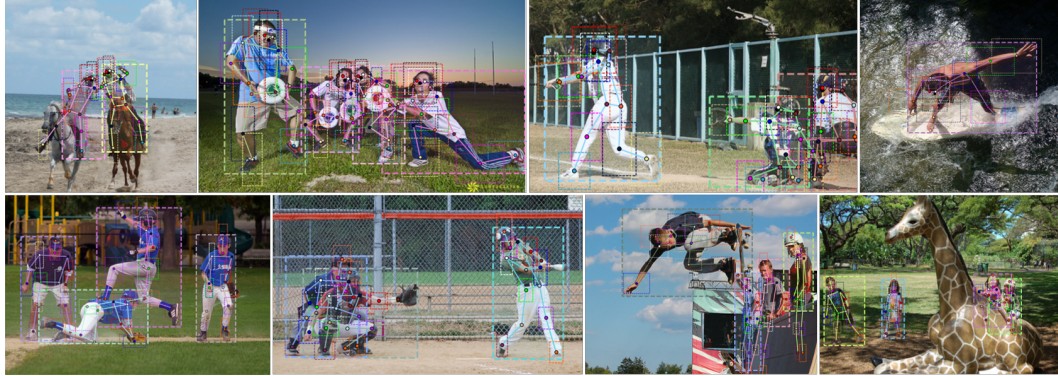

Figure 4: Qualitative results of ED-Pose on COCO (the first row) and CrowdPose (the second row). We present both explicitly detected person boxes and keypoint boxes to understand how they work.

### 5.2.2 COMPARISON OF EFFICIENCY

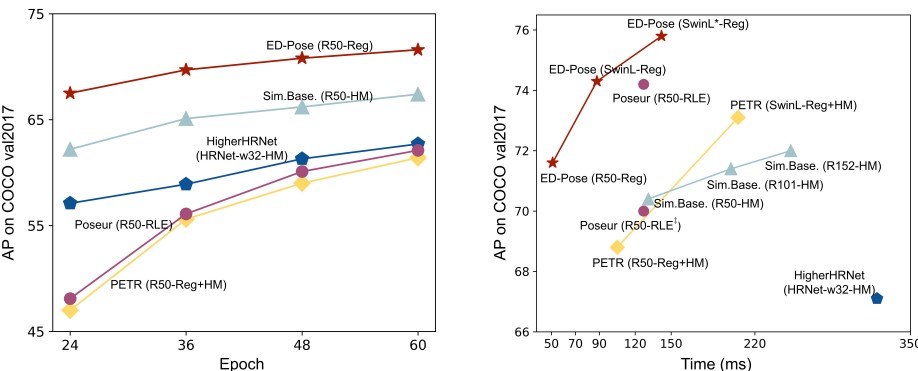

Figure 5: Comparisons of convergence speeds in the training stage (the left) and trade-offs between inference time and performance (the right) of existing mainstream methods. Our proposed one-stage method ED-Pose shows the superiority of efficiency compared with the Bottom-Up (BU) model HigherHRNet (Cheng et al., 2020), Top-Down (TD) models Sim.Base. (Xiao et al., 2018) and DETR-based Poseur (Mao et al., 2022), the one-stage method PETR (Shi et al., 2022).

For the inference time, ED-Pose surpasses all the bottom-up and one-stage methods in both speed and accuracy fields from Table. 3 and 8. Notably, from the right of Fig. 5, we compare the inference time under different AP scores. ED-Pose achieves the best efficiency and effectiveness trade-offs. Under the competitive performance (about 74.2) with the DETR-Based top-down method (Poseur with RLE loss (Li et al., 2021a)), ED-Pose can boost inference speed by 30.7%. Moreover, the left of Fig. 5 shows the convergence speeds in the training stage that are rarely discussed before. EP-Pose introducing explicit detection is faster than all methods and obtains better performance under the early epochs. Interestingly, all DETR-based methods are slower than even bottom-up methods.

### 5.3 ABLATION STUDY

**Explicit human detection**: We first analyze the effectiveness of explicit human detection on both the COCO `val2017` and the CrowdPose `test`, using the ResNet-50 backbone. We remove all human detection losses in decoder layers to verify its effectiveness. Results in Table. 4 clearly

verify that explicit human detection supervision significantly improves the convergence speed of keypoint detection and precision, yielding $+4.5$ AP on COCO and $+9.9$ AP on CrowdPose.

Table 4: Impact on explicit human detection (Human Det.) for convergence speed and precision on the COCO dataset (the left) and CrowdPose dataset (the right).

| Human Det. | 12e | 24e | 36e | 48e | 60e | | Human Det. | 12e | 24e | 36e | 48e | 60e | 80e |
|---|---|---|---|---|---|---|---|---|---|---|---|---|---|
| | 41.1 | 56.6 | 61.0 | 64.7 | 67.1 | | | 13.1 | 20.5 | 31.1 | 42.6 | 51.3 | 60.0 |
| ✓ | 60.5 | 67.5 | 69.7 | 70.8 | 71.6 | | ✓ | 37.1 | 54.8 | 62.4 | 66.4 | 68.2 | 69.9 |

**Keypoint detection representation**: ED-Pose reformulates pose estimation as the keypoint box detection, where the 2D center-point coordinate $(x, y)$ of keypoint is extended to the 4D representation $(x, y, w, h)$. As the region of feature aggregation for keypoint depends on the width and height of the keypoint box, we explore the effect of $(w, h)$ under five initialization ways: 1). **None** discards width and height and keeps the original 2D coordinate representation ; 2). **Min.** denotes that the keypoint is initialized by $1\%$ width and height of the corresponding human box; 3). **Max.** means that a keypoint is directly initialized by the width and height of its corresponding human box; 4). **FFN.** is that we apply an FFN network upon the human content query to regress a 2D weight of each keypoint within the human box ; 5). **Ours** means that we initialize learnable embeddings for each keypoint across the whole dataset and utilize it to weigh the width and height of the human box. As shown in Table. 5, the result without width and height representation is much lower than those of the other four settings, leading to a $13.3$ AP drop compared with the best initialization way (Ours) since it will lose contextual information to query the local and global relations. Learnable width and height obtain better performance than the fixed sizes.

**The interactive learning between human and keypoint detection**: Our human-to-keypoint detection decoder enhances the interactivity among external human-to-human, internal human-to-keypoint, and internal keypoint-to-keypoint. Thus, we explore the impact of different interactive learning strategies: 1) **Full** makes all human queries and keypoint queries interact one by one; 2) **w/o H-K** removes the internal human-to-keypoint interaction; 3) **w/o H-H** discards the external human-to-human interaction; 4) **Ours** is our full interactive learning strategy. As shown in Table. 6, direct full connection for all queries could lead to $1.2$ AP degradation. The reason is that external keypoint-keypoint interaction in a full connection strategy may cause a large disturbance, especially in crowded situations. Besides, other interactive strategies slightly affect performance as well.

**Hyperparameter tuning**: To explore the effect on the number $M$ of selected queries in Sec. 4.2, we conduct three numbers with $50$, $100$, and $200$, respectively. As shown in Table. 7, increasing selected human queries can improve performance, but it meets a bottleneck for further improvement.

Table 5: Impact on the width and weight initialization of the keypoint box.

| $(\mathbf{w}, \mathbf{h})$ | None | Min. | Max. | FFN. | Ours |
|---|---|---|---|---|---|
| AP | 58.3 | 70.8 | 70.8 | 71.2 | **71.6** |
| $\text{AP}_M$ | 57.5 | 65.1 | 64.9 | 65.7 | **65.9** |
| $\text{AP}_L$ | 60.0 | 79.2 | 79.3 | 79.2 | **79.8** |

Table 6: Impact on interactive learning between human and keypoint detection.

| Strategy | Full | w/o H-K | w/o H-H | Ours |
|---|---|---|---|---|
| AP | 70.4 | 71.2 | 71.3 | **71.6** |
| $\text{AP}_M$ | 64.6 | 65.6 | **66.0** | 65.9 |
| $\text{AP}_L$ | 78.8 | 79.3 | 79.0 | **79.8** |

Table 7: Impact on the number of $M$ selected queries.

| $M$ | 50 | 100 | 200 |
|---|---|---|---|
| AP | 70.9 | **71.6** | **71.6** |
| $\text{AP}_M$ | 65.2 | 65.9 | **66.0** |
| $\text{AP}_L$ | 79.2 | **79.8** | 79.6 |

## 6 CONCLUSION

In this work, we re-consider the multi-person pose estimation task as two explicit box detection processes. We unify the global person and local keypoint into the same box representation, and they can be optimized by the consistent regression loss in a fully end-to-end manner. Based on the novel methods, the proposed ED-Pose surpasses existing end-to-end methods by a large margin and shows superiority over the long-standing two-stage methods on both COCO and CrowdPose. This is a simple attempt to make the whole pipeline succinct and effective. We hope this work could inspire further one-stage designs.

## ACKNOWLEDGEMENT

The work is partially supported in by the Young Scientists Fund of the National Natural Science Foundation of China under grant No. 62106154, by the Natural Science Foundation of Guangdong Province, China (General Program) under grant No.2022A1515011524, and by Shenzhen Science and Technology Program ZDSYS20211021111415025.

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

# Appendix:
# Explicit Box Detection Unifies End-to-End Multi-Person Pose Estimation

In this Appendix, we provide descriptions of a detailed experimental setup (in Sec. A), more comparisons under COCO test-dev (in Sec. B), results of human detection on both COCO and CrowdPose datasets (in Sec. C), more visualization to show the effect of explicit box detection and comparison with PETR (in Sec. D), more discussion for ED-Pose (in Sec. E). We also append our code to reproduce the results.

## A  EXPERIMENT SETUP

**Dataset.** Our experiments are mainly conducted on the popular COCO2017 Keypoint Detection benchmark (Lin et al., 2014), which contains about $250K$ person instances with 17 keypoints. We compare with other state-of-the-art methods on both the `val2017` set and `test-dev` set. To verify the superiority of explicit detection, we also evaluate our approach on the CrowdedPose dataset (Li et al., 2019) which is more challenging and includes many crowded and occlusion scenes. It consists of $20K$ images containing about $80K$ persons with 14 keypoints. For ablation studies, we report results on the COCO `val2017` set. The OKS-based Average Precision (AP) is employed as the main evaluation metric on both datasets.

**Implementation details.** In the training stage, we augment input images by random crop, random flip, and random resize with the shorter sides in $[480, 800]$ and the longer sides less or equal to 1333 following DETR (Carion et al., 2020) and PETR (Shi et al., 2022). To accelerate the early explicit human detection, we use a human query denoising training strategy from DN-DETR (Li et al., 2022a). We use the AdamW (Kingma & Ba, 2014; Loshchilov & Hutter, 2017) optimizer with weight decay of $1 \times 10^{-4}$ and train our model on Nvidia A100 GPUs with batch size 16 for 60 epochs and 80 epochs on COCO and CrowdPose, respectively. The initial learning rate is $1 \times 10^{-4}$ and is decayed at the 55th epoch and 75th epoch by a factor of 0.1 on COCO and CrowdPose, respectively. The channel dimension $D$ is set to 256. The number of layers in Human Detection Decoder and Human-to-Keypoint Detection Decoder are 2 and 4 respectively. In the test stage, the input images are resized to have their shorter sides being 800 and their longer sides less or equal to 1333.

**Loss Function.** the overall loss function of ED-Pose can be formulated as:

$$L = L_h + L_c + L_k \tag{1}$$

$$L_h = \mu |H - H^*| + \beta(1 - \text{GIOU}) \tag{2}$$

$$L_c = -\lambda\alpha(1 - p_t)^\gamma log(p_t), \text{where } p_t = p \text{ if } y = 1,\ p_t = 1 - p \text{ if } y \neq 1 \tag{3}$$

$$L_k = \omega |P - P^*| + \theta\frac{\sum_i^K \exp(-|P_i - P_i^*|/2s^2k_i^2)\delta(v_i > 0)}{\sum_i^K \delta(v_i > 0)} \tag{4}$$

where $L_h$ is for human box regression that contains L1 loss and GIOU (Rezatofighi et al., 2019) loss, $L_c$ is for human classification that is focal loss (Lin et al., 2017) with $\alpha = 0.25$, $\gamma = 2$, and $L_k$ is for keypoint regression that includes L1 loss and the constrained L1 loss-OKS loss (Shi et al., 2022). $|H - H^*|$ is the L1 distance between the predicted human boxes and the ground-truth ones. $y \in \pm 1$ specifies the ground-truth class, and $p \in [0, 1]$ is the ED-Pose's estimated probability for the class with label $y = 1$. $|P - P^*|$ is the L1 distance between predicted keypoints inside a human and the ground-truth ones. $|P_i - P_i^*|$ is the L1 distance between the $i$-th predicted keypoint and ground-truth one, $v_i$ is the visibility flag of the ground truth, $s$ is the object scale, and $k_i$ is a per-keypoint constant that controls falloff.

The loss coefficients $\mu, \beta, \lambda, \omega, \theta$ are 5, 2, 2, 10, 4.

**The Detailed Interactive Learning between Human Detection and Keypoint Detection.** The relations from internal human-keypoint, internal keypoint-keypoint, and external human-human are learned from the self-attention mechanism. As shown in Figure. 3-(c), Human-to-Keypoint Interac-

tive Attention can be computed as follow:

$$\mathbf{Q}_{H,K}^{c'} = \text{softmax}(\frac{f(\mathbf{Q}_{H,K}^c + \text{PE}(\mathbf{Q}_{H,K}^p)) \cdot f(\mathbf{Q}_{H,K}^c + \text{PE}(\mathbf{Q}_{H,K}^p))^{\mathrm{T}}}{\sqrt{D}} + \mathcal{M}) \cdot f(\mathbf{Q}_{H,K}^c) \quad (5)$$

where PE denotes positional encoding, $f(\cdot)$ is linear projection, $\mathbf{Q}_{H,K}^c \in \mathbb{R}^{(M+M*K)\times D}$ and $\mathbf{Q}_{H,K}^p \in \mathbb{R}^{(M+M*K)\times 4}$ are human-keypoint content queries and human-keypoint position queries for $M$ human candidates and the corresponding $K$ keypoints. $D$ is 256 by default. In practice, to implement the three interactive learning processes in a simple way, we use an attention mask $\mathcal{M} \in \mathbb{R}^{(M+M*K)\times(M+M*K)}$ to block the interactiveness between external keypoint-keypoint. Thus, $\mathcal{M}$ can be formulated as:

$$\mathcal{M}(i,j) = \begin{cases} 0 & \mathcal{M}(i,j) = \text{True} \\ -\infty & \mathcal{M}(i,j) = \text{False} \end{cases} \quad (6)$$

where $\mathcal{M}(i,j)$ is the location of the attention mask. We use the $\mathcal{M}$ to keep internal human-keypoint attention (True), internal keypoint-keypoint attention (True), and external human-human attention (True) while avoiding external keypoint-keypoint attention (False).

**Inference time.** 1) Comparison Methods: All of the methods in Table. 3 are tested on our A100 machine for a fair comparison and the detector of top-down methods is tested by MMdetection. In tabel. 8, the inference time with ***emphasized bold number*** is tested by our A100 machine, otherwise, it is from PETR's paper using a V100 GPU due to no available source code. 2) Testing Rules: We omit the time for data pre-processing and only measure the time for model forwarding and data post-processing (*i.e.*, grouping operation in bottom-up methods). For bottom-up methods and one-stage methods, we set the batch size to 1. For top-down methods, we set the batch size to 5 for simulating the multi-person situation.

**A Unique Merit of ED-Pose for Human Detection Pre-training.** Thanks to the explicit introduction of human detection, ED-Pose has a unique merit compared to other existing two-stage methods and one-stage methods, which can pre-train human detection to improve the detection performance to assist the subsequent pose estimation in a fully end-to-end manner. Objects365 (Shao et al., 2019b) is a large-scale detection dataset with over 1.7M annotated images for training and 80, 000 annotated images for validation. We pre-train the human detection of ED-Pose (*i.e.*, backbone, encoder, human detection decoder) on Objects365 for 26 epochs using 64 Nvidia A100 GPUs.

## B    COMPARISON OF COCO TEST-DEV.

From Table. 8, our method without dense heatmap loss can significantly outperform all existing one-stage methods, such as DirecetPose (Tian et al., 2019a), FCPose (Mao et al., 2021b), Inspose (Shi et al., 2021), CenterNet (Zhou et al., 2019), and PETR (Shi et al., 2022). To be specific, our proposed ED-Pose has 2.2 AP gains compared with the fully end-to-end method PETR (Shi et al., 2022) with a ResNet-50 backbone or Swin-L backbone with about $50\%$ inference time reduction.

## C    RESULTS OF HUMAN DETECTION

Due to the explicit person detection in our methods, we also report the human detection performance on the COCO `val2017` set and CrowdPose `test` set in Figure. 9. For the COCO dataset, ED-Pose with Swin-L* backbone gains 7.7 $\text{AP}_M$ and 6.7 $\text{AP}_L$ improvement compared with Faster-RCNN (Ren et al., 2015). For the CrowdPose dataset, most top-down methods directly use YOLOv3 (Redmon & Farhadi, 2018) pre-trained on COCO `trainval` to generate the detected human bounding boxes, making their performance even worse than recent bottom-up methods. ED-Pose can provide human detection results with nearly doubling performance on $\text{AP}_M$ and $\text{AP}_L$ to serve future work. Moreover, in Table 10, we study the impact of different decoders (*i.e.*, human detection decoder and human-keypoint detection decoder) for human detection and keypoint detection. First, we discover that the keypoint initialization provided by the human detection decoder has already achieved good results on COCO and CrowdPose datasets, which proves the effectiveness of explicit human detection. Second, the keypoint detection introduced via the human-keypoint detection decoder is also helpful to improve large-scale human detection and obtain similar performance

Table 8: Comparisons with state-of-the-art one-stage methods on COCO `test-dev` dataset. underlined highlights the compared results. *Emphasized bold number* in inference time is tested by our A100 machine.

| | Method | Backbone | Loss | AP | $AP_{50}$ | $AP_{75}$ | $AP_M$ | $AP_L$ | Time [ms] |
|---|---|---|---|---|---|---|---|---|---|
| Non E-2-E | DirectPose | ResNet-50 | R. | 62.2 | 86.4 | 68.2 | 56.7 | 69.8 | 74 |
| | DirectPose | ResNet-101 | R. | 63.3 | 86.7 | 69.4 | 57.8 | 71.2 | - |
| | FCPose | ResNet-50 | R+HM. | 64.3 | 87.3 | 71.0 | 61.6 | 70.5 | 68 |
| | FCPose | ResNet-101 | R+HM. | 65.6 | 87.9 | 72.6 | 62.1 | 72.3 | 93 |
| | InsPose | ResNet-50 | R+HM. | 65.4 | 88.9 | 71.7 | 60.2 | 72.7 | *78* |
| | InsPose | ResNet-101 | R+HM. | 66.3 | 89.2 | 73.0 | 61.2 | 73.9 | 100 |
| | CenterNet | Hourglass | R. | 63.0 | 86.8 | 69.6 | 58.9 | 70.4 | 160 |
| Fully E-2-E | PETR | ResNet-50 | R+HM. | 67.6 | 89.8 | 75.3 | 61.6 | 76.0 | *105* |
| | PETR | Swin-L | R+HM. | 70.5 | 91.5 | 78.7 | 65.2 | 78.0 | *206* |
| | ED-Pose | ResNet-50 | R. | 69.8$\uparrow_{2.2}$ | 90.2 | 77.2 | 64.3 | 77.4 | *51*$\downarrow_{51.4\%}$ |
| | ED-Pose | Swin-L | R. | **72.7**$\uparrow_{2.2}$ | **92.3** | **80.9** | **67.6** | **80.0** | *88*$\downarrow_{57.3\%}$ |

[1] The inference time without *Emphasized bold number* is from PETR's paper because there is no public code for replication.

on medium-scale human detection. Different from previous observations of severe optimization conflicts between human detection and keypoint detection, ED-Pose unifies the contextual learning between human-level and keypoint-level information and gains benefits from each other.

Table 9: The Average Precision comparisons of the commonly used human detection methods (e.g., Faster-RCNN and YOLOv3) with ours on COCO (the left) and CrowdPose (the right). Notably, We only compare $AP_M$ (medium object) and $AP_L$ (large object) here as small objects in the two datasets are not labeled with keypoints. ED-Pose can provide better human detection to serve future work, especially for top-down and one-stage methods

| Methods | $AP_M$ | $AP_L$ | Methods | $AP_M$ | $AP_L$ |
|---|---|---|---|---|---|
| Faster-RCNN | 63.3 | 74.5 | YOLOv3 | 37.4 | 46.5 |
| ED-Pose (ResNet-50) | 65.9 | 77.6 | ED-Pose (ResNet-50) | 58.0 | 75.3 |
| ED-Pose (Swin-L) | 69.7 | 80.2 | ED-Pose (Swin-L) | 61.8 | 78.4 |
| ED-Pose (Swin-L*) | **71.0** | **81.2** | ED-Pose (Swin-L*) | **66.8** | **82.2** |

Table 10: The performance comparisons of results from different decoders (*i.e.*, human detection decoder and human-keypoint detection decoder) on both COCO and CrowdPose datasets with the Swin-L as the backbone.

| Method | Dataset | Decoder | $AP_M$ (Human) | $AP_L$ (Human) | AP (Keypoint) |
|---|---|---|---|---|---|
| ED-Pose | COCO | Human Det. | **70.4** | 79.6 | 70.4 |
| ED-Pose | COCO | Human Det.+Human-Keypoint Det. | 69.8 | **80.1** | **74.3** |
| ED-Pose | CrowdPose | Human Det. | **61.7** | 76.9 | 68.0 |
| ED-Pose | CrowdPose | Human Det.+Human-Keypoint Det. | 61.6 | **78.4** | **73.1** |

# D  QUALITATIVE RESULTS

**Qualitative ablations for explicit human detection**: In the main article, we verify the effectiveness of explicit human detection for convergence speed and precision of the keypoint detection. Here, we present the corresponding qualitative results on the CrowdPose `test` set as shown in Fig. 6. In general, explicit human detection has several advantages: 1). It can provide global human-level information, enabling the model to be aware of flipping easily like the case in the first row. 2). It can give a clear human identity to the crowded scene to avoid the keypoint mismatching across persons, such as the case in the second row. 3) It is friendly for small human pose estimation as it can provide a precise prior of human box position, e.g., the case in the third row.

**Qualitative comparisons between PETR and ED-Pose**: We present the visualization comparisons of ED-Pose and PETR (Shi et al., 2022) on the COCO dataset, as shown in Fig 7. From the first row, ED-Pose can pay attention to the flipping issue due to the introduction of explicit detection and the realization of human-keypoint feature propagation. The second row further reflects that explicit detection in ED-Pose can make the model aware of the human position when conducting pose estimation, rather than the chaotic and unconscious query on a full image like PETR. All

possible detected keypoints from PETR have low scores, making it hard to distinguish the right and wrong results. The third row shows that the enhancement of the global-local interactivity could relieve the estimation problem of the hard pose with heavy occlusions.

# E    MORE DISCUSSION

**Discussion for Additional Related Works.** (1) N. xue (Xue et al., 2022) focuses on improving the performance of the heatmap-based bottom-up method by learning the local-global contextual adaptation without any explicit box detection. (2) Different from the one-stage method, D. Wang (Wang & Zhang, 2022) decouples persons into multiple instance-aware feature maps to obtain the keypoints without any explicit box detection, and they also use heatmap-based supervision.

Different from their works, ED-Pose aims to utilize two explicit box detection processes with a unified box representation and light L1 losses. It abandons all post-processings in a fully end-to-end manner. In terms of performance, ED-Pose surpasses them by a margin and even outperforms heatmap-based Top-down methods under the same backbone.

**The More Detailed Advantage of ED-Pose.** From the results, ED-Pose is efficient and effective even with the same backbone compared with previous works, indicating it really helps with pose estimation. From the method with two box detection processes, we have admitted it is inspired by recent DETR-based methods. However, the motivation and usage have obvious differences due to the task differences between object detection and multi-person pose estimation. Multi-person pose estimation focuses on either human-level (global) or keypoint-level (local) information. ED-Pose re-considers this task as two explicit box detection processes with a unified representation and regression supervision. Moreover, we also compare with other DETR-based pose estimators, and ED-Pose shows its superiorities in both efficiency and performance. The first human box detection can extract global features and provide a good initialization for the latter keypoint detection, making the training process converge fast. The second keypoint box detection can bring in local contextual information near keypoints. However, previous DETR-based pose estimation methods only directly regressed the keypoints' 2D coordinates, we argue that the keypoint representation as a point lacks local contextual information. They directly regress local keypoints without global feature extraction, making the convergence slow and inaccurate estimation. In Figure. 6 and Figure. 7, we demonstrate the qualitative comparisons of whether or not we needed to use explicit detection and PETR with ours. Lastly, we conduct comprehensive experiments on how explicit detection works in Table. 4 and Table. 5.

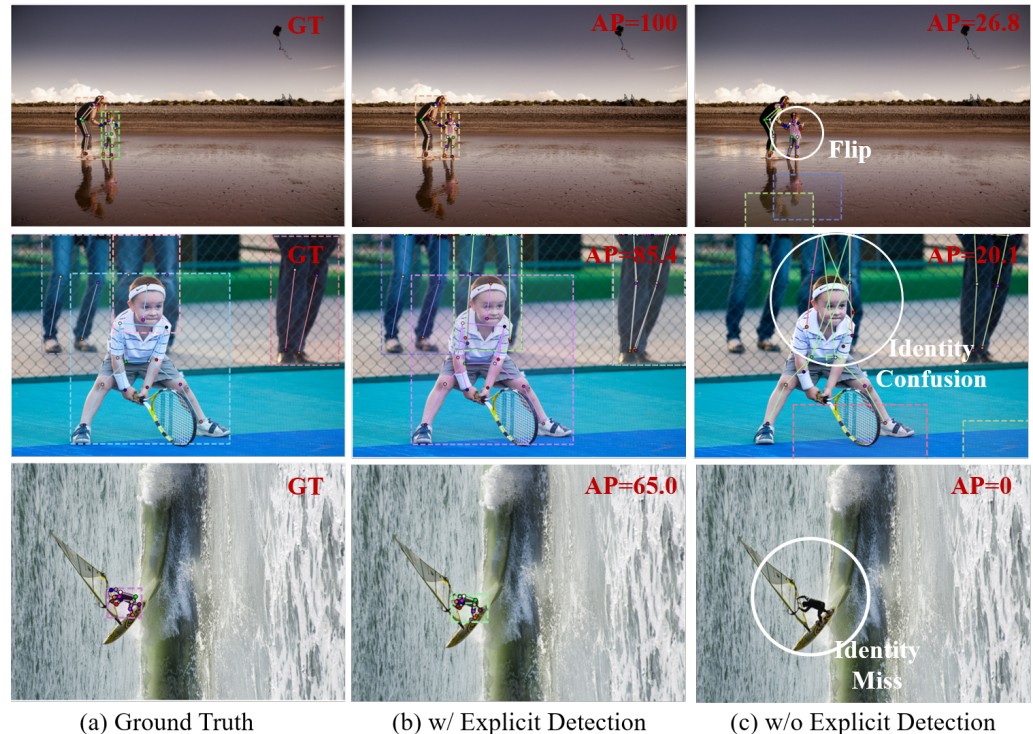

Figure 6: Visualization results of the ablation study for explicit human detection on the CrowdPose dataset. Notably, we ignore the keypoint box to compare the keypoint localization of the two settings clearly. The white circle is the failure part with a description of the type of failure.

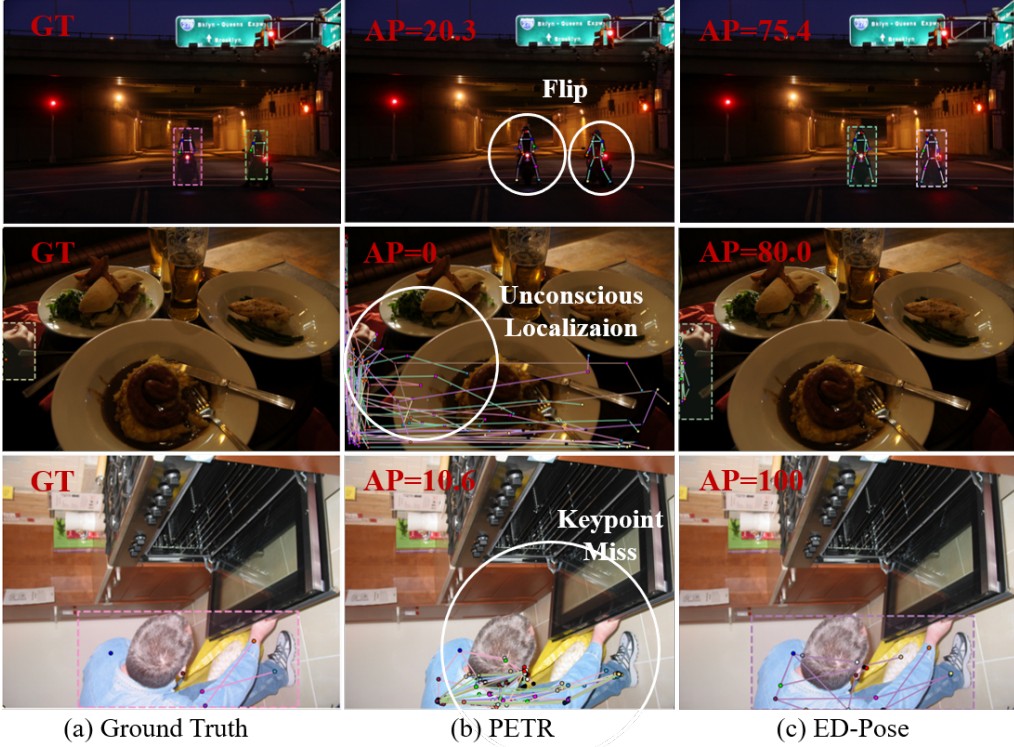

Figure 7: Visualization results of PETR and ED-Pose on COCO dataset. Both methods are based on ResNet-50 as the backbone. Notably, there are several candidates in the PETR's result at the second line as their classification scores are close and relatively low.

