# OpenReview forum: "Explicit Box Detection Unifies End-to-End Multi-Person Pose Estimation"
_ICLR.cc/2023/Conference — ICLR 2023 poster_

### Official Review · Reviewer_EVzH · 2022-10-23

**Confidence:** 3
**Correctness:** 4
**Technical Novelty And Significance:** 3
**Empirical Novelty And Significance:** Not applicable
**Recommendation:** 8

**Clarity, Quality, Novelty And Reproducibility:**

The paper is well writen and easy to follow. The proposed mehtod is novel and effective.

**Strength And Weaknesses:**

Strengths:
1. The proposed approach introduces human tokens to interact with keypoint tokens to resolve the ambiguous association between keypoints and humans in the transformer framework.The approach is reasonable and novel. Explicit human detection contributes significant performance improvement in the proposed framework. The interaction between human and keypoint tokens further improves the performance.

2. The representation of (x, y, w, h) instead of (x, y) for keypoints provides more contextual information for keypoint estimation as demonstrated in Table 5.

3. The proposed approach achieves state-of-the-art performance on COCO and CrowdPose datasets with competing inference speed as shown in Tables 2 and 3. The proposed approach also has faster convergence speed than competing methods as shown in Figure 5.

Weaknesses:
Some implementation details are missing:
(1) The detailed losses are not presented in neither the paper nor the supplementary material.
(2) Some details of the network structures are not provided (e.g. how many human detection decoders and human-to-keypoint detection decoders are used?)


**Summary Of The Paper:**

The paper presents an end-to-end approach for multi-person pose estimation. It unifies human detection and keypoint estimation in an one-stage framework. Based on transformer, human tokens and keypoint tokens are introduced and interacted to generate discriminate representations for human detection and keypoint estimation. The proposed aporoach achives state-of-the-art performance on COCO and CrowdPose datasets. Extensive experiments demonstrate the effectiveness of the the components in the proposed approach.

**Summary Of The Review:**

Overall, the paper is well written. The novelty and contributions of the paper are sufficient.

---

> ### Author Response · Authors · 2022-11-15
> **Rebuttal by Paper577 Authors**
>
> 1. **The detailed losses are not presented in either the paper or the supplementary material.**
>
> We thank you for pointing it out! We have added it to our Appendix. Formally, the overall loss function of ED-Pose can be formulated as:
>        $$L=L\_h+L\_c+L\_k $$
>        $$L\_h=\mu \left|H-H^\*\right|+\beta(1-\mathrm{GIOU})$$
>        $$L\_c=- \lambda \alpha(1-p\_t)^{\gamma}log(p\_t) \quad \mathrm{where} \quad p_t=p \quad \mathrm{if} \quad y=1, \quad p_t=1-p \quad \mathrm{if}\quad  y\neq1$$
>        $$L\_k=\omega \left|P-P^\*\right| + \theta \frac{\sum^{K}\_i\mathrm{exp}(-\left|P\_i-P\_i^\*\right|/2s^2k\_i^2)\delta(v\_i>0)}{\sum^{K}\_i\delta(v\_i>0)}$$
>
> where $L\_h$ is for human box regression that contains L1 loss and GIOU loss, $L\_c$ is for human classification that is focal loss with $\alpha$ = 0.25, $\gamma$ = 2, and $L\_k$ is for keypoint regression that includes L1 loss and the constrained L1 loss-OKS loss. $\left|H-H^*\right|$ is the L1 distance between the predicted human boxes and the ground-truth ones. $y\in{\pm1}$ specifies the ground-truth class, and $p\in[0, 1]$ is the ED-Pose's estimated probability for the class with label $y=1$. $\left|P-P^\*\right|$ is the L1 distance between predicted keypoints inside a human and the ground-truth ones. $\left|P\_i-P\_i^\*\right|$ is the L1 distance between the $i$th predicted keypoint and ground-truth one, $v\_i$ is the visibility flag of the ground truth, $s$ is the object scale, and $k\_i$ is a per-keypoint constant that controls falloff.
>
> The loss coefficients $\mu, \beta, \lambda, \omega, \theta$ are $5$, $2$, $2$, $10$, $4$.
>
> 2. **Some details of the network structures are not provided (e.g. how many human detection decoders and human-to-keypoint detection decoders are used?)**
>
> As illustrated in Figure 3, Human Detection Decoder has $2$ layers, and the Human-to-Keypoint Detection Decoder has $4$ layers. We also have added it to our Appendix.

---

### Official Review · Reviewer_Dxnp · 2022-10-23

**Confidence:** 4
**Clarity, Quality, Novelty And Reproducibility:** This paper is written clearly. Novelt…
**Correctness:** 4
**Technical Novelty And Significance:** 3
**Empirical Novelty And Significance:** 3
**Recommendation:** 6

**Strength And Weaknesses:**

Strength:
1. Paper is written clearly.
2. Global and local information are  reasonably combined by query initialization and human-to-keypoint query expansion.
3. Human detection and keypoint detection are integrated into an end-to-end and succinct framework. each part of this framework can benefit each other during training.
4. The proposed unified framework achieves promising accuracy and good efficiency.

Weaknesses:
The whole model architecture can be considered as the variant of  DETR

**Summary Of The Paper:**

This paper proposes a method for multi-person pose estimation method, which integrates human detection and human pose estimation into an end-to-end framework. The first part of this framework is a human detector based on deformable DETR and DAB-DETR, and outputs human-instance features and estimated human bounding boxes.  The second part, i.e. keypoint detector, is fed with a human-instance query and a set of keypoint queries which are initialized with its corresponding human-instance query. By doing so, the queries of keypoints belonging to one human instance can interact with each other while each keypoint query can also interact with its human-instance query. The keypoint locations are regressed by the keypoint detector. Experiments are performed on multiple datasets. the proposed method significantly outperforms one-stage SOTAs. When compared with two-stage methods, the proposed method performs better than bottom-up SOTAs, while performs comparably or worse than top-down SOTAs but with better efficiency and more succinct pipeline.  Besides, the ablation study also verifies the effectiveness of some proposed algorithm components.

**Summary Of The Review:**

The proposed one-stage end-to-end framework for multi-person pose estimation is succinct and efficient. The accuracy is also promising. The effective combinations of human detection and keypoint detection, global and local information are a novel extension of existing DETR architectures.

---

> ### Author Response · Authors · 2022-11-15
> **Rebuttal by Paper577 Authors**
>
> **Weaknesses: The whole model architecture can be considered as the variant of DETR**
>
> Thanks for your kind feedback! We admit this work is inspired by DETR-based methods, which do not require post-processing and could achieve full end-to-end training for multiple object detection.
> However, it could not be regarded as a weakness simply because of the variant. This work has many unique contributions to our interested multi-person pose estimation, such as the introduction of two explicit box detection processes for both human-level (global) and keypoint-level (local) content learning, unifying keypoint representation into boxes under lightweight regression supervision, interactive learning strategy to enhance the global and local feature aggregation. Generally speaking, it is efficient and effective.
> DETR is presented for object detection, where it only learns to regress the position of the human in the image while does not capture the posture information inside the person. For the task of multi-person pose estimation, we need to extract the global person features and local keypoint information efficiently. ED-Pose unifies the contextual learning between human-level (global) and keypoint-level (local) information.
> Different from the previous one-stage method--PETR (CVPR'22) that is also built on DETR, ED-Pose re-considers this task as two explicit box detection processes with a unified representation and regression supervision. It is worth mentioning that, for the first time, as a fully end-to-end framework with an L1 regression loss, ED-Pose surpasses heatmap-based Top-down methods under the same backbone by 1.2 AP on COCO and achieves the state-of-the-art with 76.6 AP on CrowdPose without bells and whistles.
> Moreover, we first address the problem of the slow convergence speed in DETR-based pose estimation methods, which has not been discussed in previous papers.

---

### Official Review · Reviewer_h7oS · 2022-10-26

**Confidence:** 3
**Correctness:** 3
**Technical Novelty And Significance:** 2
**Empirical Novelty And Significance:** 3
**Recommendation:** 6

**Clarity, Quality, Novelty And Reproducibility:**

I think the reproducibility of this paper is quite good as the authors provide their code. For the others, I still have some concern on the clarity and the quality (see the Strength And Weaknesses for more details).

**Strength And Weaknesses:**

Strength:
1. The paper gives a clear summary and analysis of existing methods, which motivates their method well.
2. Many experiments have been conducted and the results seem satisfactory.

Weakness:
1. My main concern with this paper is the proposed explicit box detection process. Specifically, I think a clearer definition of what is the so-called explicit box detection process should be given more precisely. In the current version, it seems that I can only guess what it means, which brings me reading difficulties. From my guess, does this explicit box detection process just mean to explicitly detect bounding boxes for both the human and the keypoints? I personally use this understanding to understand this paper and I am sorry if I misunderstand it.
2. Then with such an understanding, I think the advantage of such a process should be mentioned in more detail. Only demonstrating the advantage of such a process through experiments seems to be not enough to me. Besides, I am not sure if this process has improved the results because it really helps with pose estimation, or because it is more relevant to the task of object detection and therefore fits DETR and other existing object detection methods better. Can the authors also explain this?
3. I also have some other minor worries w.r.t. the experiments. (1) I am not sure whether the authors overclaim their improvement a bit as it seems a large proportion of their improvement on Swin-L (1.5 -> 5.0 on crowdpose and 1.2 -> 2.7 on coco) is because they use an extra dataset Objects365. (2) I am not sure if it is allowed to put the result on COCO-test dev in the appendix only, as I am not sure whether this should be considered as a new dataset or not.

**Summary Of The Paper:**

In summary, to deal with the task of multi-person pose estimation, this paper regards the task as two explicit box detection processes. This enables the model to be optimized in an end-to-end manner and achieves good result.

**Summary Of The Review:**

Overall, my main concern w.r.t. this submission is on the proposed explicit box detection process (see the Strength And Weaknesses for more details). Specifically, I think there are some things that should be made clearer before the acceptance of this submission.

---

> ### Author Response · Authors · 2022-11-15
> **Rebuttal by Paper577 Authors**
>
> 1. **Does this explicit box detection process mean to explicitly detect bounding boxes for both the human and the keypoints?**
>
> Yes. The explicit box detection process is to detect bounding boxes for both the human and the keypoints.
>
> 2. **The advantage of such a process should be mentioned in more detail.**
>
> Thanks for your kind feedback! We have added it to our Appendix.
>
> From the results, ED-Pose is efficient and effective even with the same backbone compared with previous works, indicating it really helps with pose estimation.
> From the method with two box detection processes, we have admitted it is inspired by recent DETR-based methods. However, the motivation and usage have obvious differences due to the task differences between object detection and multi-person pose estimation. Multi-person pose estimation focuses on either human-level (global) or keypoint-level (local) information. ED-Pose re-considers this task as two explicit box detection processes with a unified representation and regression supervision.
> Moreover, we also compare with other DETR-based pose estimators, and ED-Pose shows its superiorities in both efficiency and performance. The first human box detection can extract global features and provide a good initialization for the latter keypoint detection, making the training process converge fast. The second keypoint box detection can bring in local contextual information near keypoints. However,
> previous DETR-based pose estimation methods only directly regressed the keypoints' 2D coordinates, we argue that the keypoint representation as a point lacks local contextual information. They directly regress local keypoints without global feature extraction, making the convergence slow and inaccurate estimation. In our Appendix (Fig.6 and Fig.7), we demonstrate the qualitative comparisons of whether or not we needed to use explicit detection and PETR with ours. Lastly, we conduct comprehensive experiments on how explicit detection works in Tab.4 and Tab.10.
>
> 3. **The claim of performance improvement.**
>
> Actually, we do not overclaim our method's improvement. Under the ResNet-50, we surpass the PETR, which is also the variant of the Deformable DETR, by a $2.8$ AP on the COCO dataset. Furthermore, we verify that our method still has consistent improvement under the same large backbone by a $1.2$ AP.
>
> Using the additional Objects365 dataset to pre-train human detection is a unique merit of our method compared to other one-stage methods and bottom-up methods. We show that using extra detection datasets can improve our human detection performance, and can further help the pose estimation task. This is also what we want to claim that introducing explicit human detection in a fully end-to-end trainable framework can bring huge benefits to the accuracy and convergence speed of the pose estimation task.
>
> 4. **The result on the COCO-test dev in the appendix.**
>
> No, COCO-test dev is not a new dataset. COCO provides two evaluation sets val2017 and test-dev, where the results of the test-dev set are submitted to COCO's Keypoint Leaderboard for evaluation. In the field of pose estimation, we usually train a model on the COCO2017 train set and report one of the val or test-dev results in the main article, and leave another set in the Appendix.

---

> ### Comment · Reviewer_h7oS · 2022-11-17
> **Response to Authors**
>
> Thanks to the authors for their responses. As their responses address most of my concerns, I decide to raise my rate from 5 to 6. However, I still would like to point out that although it's not a big deal, from what I know about HPE, especially 2D-HPE, putting one of the COCO val set and test dev set in the main paper and the other one in the supplementary material is at least not a "usual" convention.

---

> > ### Author Response · Authors · 2022-11-18
> > **Thanks for Your Recognition**
> >
> > Dear Reviewer #h7oS,
> >
> > We are pleased that we addressed your concerns about our paper, and we appreciate that you raised your rate to recognition our work.
> >
> > Best!
> >
> > The authors of Paper577

---

### Official Review · Reviewer_uhwo · 2022-10-28

**Confidence:** 5
**Correctness:** 4
**Technical Novelty And Significance:** 3
**Empirical Novelty And Significance:** 3
**Recommendation:** 8

**Clarity, Quality, Novelty And Reproducibility:**

Overall, the paper is well written and easy to follow.  The advantage of the proposed method is well shown. The experiments are thorough.  The code is provided (but not tested by the reviewer).

**Strength And Weaknesses:**

+ The proposed method takes advantage of the fully end-to-end DETR framework for pose estimation in a straightforward yet effective way.
+ The proposed human-to-keypoint decoder  with the box-based keypoint formulation seems to be effective.
+ The proposed method obtains strong performance on two challenging datasets.

- Some details of the box-based keypoint formulation are missing. How are those bounding boxes determined for different keypoints of different persons in training to compute the keypoint box regression loss, L_k^l? Is the same method used in both the COCO and CrowdPose datasets? If not, how is the dataset prior (e.g., crowd vs not-so crowd) leveraged?  And, how sensitive is the proposed method to the keypoint boxes?

- The interactive learning part is not clearly presented. How is the human-to-human interaction computed, as well as the human-keypoint and keypoint-keypoint?

- Two references that seem to be relevant are not discussed.  1) N. Xue et al Learning Local-Global Contextual Adaptation for Multi-Person Pose Estimation, CVPR22; 2) D. Wang and S. Zhang, Contextual Instance Decoupling for Robust Multi-Person Pose Estimation, CVPR22.

- It may be interesting to test the COCO or CrowdPose trained model on the OCHuman dataset to see the transferability as done in 1) N. Xue's paper. It will also enable better understanding the impacts of the selection of the keypoint boxes.


**Summary Of The Paper:**

This paper presents a method of casting the problem of multi-person pose estimation as a two-level (human and keypoints) Explicit-box Detection problem. It is built on the DETR framework and its variants such as the Deformable DETR and DN-DETR.  The human detection component is a direct application of the existing DETR variant. The proposed human-to-keypoint decoder with the box-based keypoint formulation is the novel component. The proposed method is end-to-end trainable without resorting to post-processing, thanks to the DETR framework. It obtains better accuracy performance than the prior art on the COCO keypoint benchmark and the CrowdPose benchmark.

**Summary Of The Review:**

Overall, this is a good paper on multi-person pose estimation. It leverages the strong DETR framework with a new human-to-keypoint component.  The reviewer would like to see the rebuttal on the aforementioned weaknesses.

---

> ### Author Response · Authors · 2022-11-15
> **Rebuttal by Paper 577 Authors (Part 1)**
>
> 1. **How are those bounding boxes determined for different keypoints of different persons in training to compute the keypoint box regression loss, $L_k^l$**?
>
> We thank the reviewer for pointing out the ambiguous descriptions in the original paper.
>
> The keypoint box regression loss, $L\_k^l$ can be formulated as follow:
>
> $$L\_k^l=\left|P-P^*\right|$$
>
> where $P$ is the $2$D coordinate $(x,y)$ of the predicted keypoint box and $P^*$ is the ground-truth one. In $L\_k^l$, we only supervise the center of the keypoint box because all of the datasets are only labeled with $2$D keypoint. However, the width and height of the keypoint box are influenced by the corresponding human box's size and implicitly supervised in the Human-to-Keypoint Interactive Attention layer and Deformable Token-to-Keypoint Attention layer in Figure.3-(b).
>
> In detail, we explain the initialization and update process of the keypoint box step-by-step.
>
> 1). After Human Detection Decoder, we can obtain $M$ candidate human boxes (human positional queries $\mathbf{Q}\_{H\_s}^p\in \mathbb{R}^{M\times 4}$),
> and their corresponding feature information (human content queries $\mathbf{Q}\_{H\_s}^c \in \mathbb{R}^{M \times 256}$).
>
> 2). We randomly initialize $K$ learnable keypoint embeddings ($\mathbf{V}_e\in \mathbb{R}^{K\times 256}$), where $K$ is the number of keypoints in a specific dataset (e.g., $17$ for COCO, $14$ for CrowdPose).
>
> 3). For $M$ human candidates, we add each human feature information (each human content query  $\mathbb{R}^{1\times 256}$) and the corresponding $K$ learnable keypoint embeddings ($\mathbb{R}^{K\times 256}$) by a broadcast mechanism, to update $K$ keypoints' feature information ($\mathbb{R}^{1\times K \times 256}$) within each human. Finally, we can obtain $M$ human's keypoints's feature information ($\mathbf{Q}_{K}^{c} \in \mathbb{R}^{M\times K \times 256}$)
>
> 4). We put the initialized $K$ keypoints' feature information of $M$ human candidates ($\mathbf{Q}_{K}^{c}\in \mathbb{R}^{M\times K \times 256}$) into FFNs to regress their 2D keypoint coordinate ($ \mathbb{R}^{M\times K \times 2}$).
>
> 5). We define the keypoint as the box representation. The center of keypoint box is the 2D coordinate, and the width and height of the keypoint box are related to the size of the human box where it is located. Specifically, we randomly initialize $K$ learnable weights ($\mathbf{W}\in\mathbb{R}^{K\times 2}$). For $M$ human candidates, the width and height of $K$ keypoint boxes within each human are conditioned by the width and height of the corresponding human box ($\mathbb{R}^{1\times 2}$) via dot-multiplying the learnable weights ($\mathbf{W}\in \mathbb{R}^{K\times 2}$). Finally, we can obtain the initialized keypoint boxes of $M$ human candidates ($\mathbf{Q}_{K}^{p}\in\mathbb{R}^{M\times K \times 4}$).
>
> 6). In the Human-to-Keypoint Detection Decoder, the center, width, and height of the keypoint box are all refined layer-by-layer.
>
>
> 2. **Is the same method used in both the COCO and CrowdPose datasets? If not, how is the dataset prior (e.g., crowd vs not-so-crowd) leveraged?**
>
> Yes, our method is the same for COCO and CrowdPose. The only difference is the keypoint number $K$ ($17$ for COCO, $14$ for CrowdPose) according to their keypoint definition. In our method, another dataset prior is the number of human queries in the Human-to-Keypoint Detection Decoder, which should be larger than the maximum number of annotated humans in all images. Thus, we use $M=100$ candidate human boxes for the two datasets to satisfy the constraint.
>
> 3. **How sensitive is the proposed method to the keypoint boxes?**
>
> This is a good question. In Tab.5 (the ablation of Keypoint detection representation), we have studied the effect of the width and height initialization of the keypoint box, which shows that our proposed method can bring in a 0.8 AP improvement on the COCO dataset compared with a fixed scale. Additionally, we can observe in Figure.4 that the learnable keypoint boxes are reasonable and robust for both COCO and Crowdpose datasets.

---

> > ### Author Response · Authors · 2022-11-15
> > **Rebuttal by Paper577 Authors (Part 2)**
> >
> > 4. **How is the human-to-human interaction computed, as well as the human-keypoint and keypoint-keypoint?**
> >
> > Thanks for the feedback! We have made it clear in our Appendix.
> >
> > The relations from internal human-keypoint, internal keypoint-keypoint, and external human-human are learned from the self-attention mechanism.
> >
> > As shown in Figure.~3-(c), Human-to-Keypoint Interactive Attention can be computed as follow:
> >
> > $     \mathbf{Q}\_{H,K}^{c'}=\sigma(\frac{f(\mathbf{Q}\_{H,K}^c+\mathrm{PE}(\mathbf{Q}\_{H,K}^p))\cdot f(\mathbf{Q}\_{H,K}^c+\mathrm{PE}(\mathbf{Q}\_{H,K}^p))^\mathrm{T}}{\sqrt{D}}+\mathcal{M})\cdot f(\mathbf{Q}\_{H,K}^c)$
> >
> > where $\rm{PE}$ denotes positional encoding, $f(\cdot)$ is linear projection, $\sigma$ is softmax operation,
> > $\mathbf{Q}\_{H,K}^c \in \mathbb{R}^{(M+M\*K)\times D}$ and
> > $\mathbf{Q}\_{H,K}^p \in \mathbb{R}^{(M+M\*K)\times4}$ are human-keypoint content queries and human-keypoint position queries for $M$ human candidates and the corresponding $K$ keypoints. $D$ is 256 by default. In practice, to implement the three interactive learning processes in a simple way, we use an attention mask  $\mathcal{M} \in \mathbb{R}^{(M+M\*K) \times (M+M\*K)}$ to block the interactiveness between external keypoint-keypoint. Thus, $\mathcal{M}$ can be formulated as:
> >
> > $\mathcal{M}(i,j)= 0, \mathcal{M}(i,j)=\mathrm{True}$
> >
> > $\mathcal{M}(i,j)= -\infty, \mathcal{M}(i,j)=\mathrm{False}$
> >
> > where $\mathcal{M}(i,j)$ is the location of the attention mask. We use the $\mathcal{M}$ to keep internal human-keypoint attention (True), internal keypoint-keypoint attention (True), and external human-human attention (True) while avoiding external keypoint-keypoint attention (False).
> >
> > 5. **Two references that seem to be relevant are not discussed. 1) N. Xue et al Learning Local-Global Contextual Adaptation for Multi-Person Pose Estimation, CVPR22; 2) D. Wang and S. Zhang, Contextual Instance Decoupling for Robust Multi-Person Pose Estimation, CVPR22.**
> >
> > Thank you for pointing it out, we have added it to our Appendix. However, they have clear differences in this work.
> >
> > (1) N. xue's paper focuses on improving the performance of the heatmap-based bottom-up method by learning the local-global contextual adaptation without any explicit box detection.
> > (2) Different from the one-stage method, D. Wang's paper decouples persons into multiple instance-aware feature maps to obtain the keypoints without any explicit box detection, and they also use heatmap-based supervision.
> >
> > Different from their works, ED-Pose aims to utilize two explicit box detection processes with a unified box representation and light L1 losses. It abandons all post-processings in a fully end-to-end manner. In terms of performance, ED-Pose surpasses them by a margin and even outperforms heatmap-based Top-down methods under the same backbone.

---

> > > ### Author Response · Authors · 2022-11-15
> > > **Rebuttal by Paper577 Authors (Part 3)**
> > >
> > > 6. **It may be interesting to test the COCO or CrowdPose trained model on the OCHuman dataset to see the transferability as done in 1) N. Xue's paper. It will also enable a better understanding of the impacts of the selection of the keypoint boxes.**
> > >
> > > Thanks for your great suggestion! We share our findings as follows.
> > > We follow N. Xue's paper directly to evaluate the COCO-trained ED-Pose with different backbones on the OCHuman test set. Interestingly, as shown in the below table, our models have lower AP while higher AR than N. Xue's results, where ED-Pose with Swin-L* backbone achieves 87.7\% AR while obtaining 30.4\% AP. This is an inconsistent phenomenon as AR doesn't depend on false positives. Accordingly, we carefully visualize and double-check the ground truth of the OCHuman dataset and find that OCHuman only annotates 1-2 humans, where they have MaxIoU$>$0.5, in crowded situations even though there are more humans in the image. Furthermore, we visualize ED-Pose's results and discover ED-Pose can not only output the predictions of the annotated human but also cover the predictions of all the humans in the image. Finally, we find that the annotated individual in a crowded situation has relatively low predicted classification scores compared to other unlabelled humans, which explains why ED-Pose obtains a low AP while accurately localizing the keypoints. Some qualitative results are shown in the anonymous link \url{https://github.com/EDPose/Viz-for-OCHuman}. It is worth exploring how to use our ED-Pose model trained on COCO to serve better annotations for OCHuman.
> > >
> > > Furthermore, we use the OCHuman val set to finetune our model trained on COCO and evaluate the finetuned model by the OCHuman test set. The performance of our ED-Pose with Swin-L* backbone achieves the state-of-the-art result of 67.3 mAP as the finetune process makes our model fit the annotation criteria of the OCHuman dataset. Besides, those models have similar $AR_{50}$, indicating a similar recall capability.
> > >
> > > | Method           | Loss |   ${\rm AP}$  | ${\rm AP}_{50}$ | ${\rm AP}_{75}$ |   ${\rm AR}$  | ${\rm AR}_{50}$ | ${\rm AR}_{75}$ |
> > > |------------------------------|------|:-------------:|:---------------:|:---------------:|:-------------:|:---------------:|:---------------:|
> > > | N. Xue (HRNet-W32)           | HM   |      38.1     |       50.6      |       42.1      |      74.5     |       93.8      |       80.5      |
> > > | ED-Pose-Test (ResNet-50)     | R.   |      33.0     |       41.4      |       36.7      |      84.8     |       97.5      |       91.1      |
> > > | ED-Pose-Finetune (ResNet-50) | R.   |      65.0     |       73.2      |       69.2      |      91.1     |       98.0      |       95.1      |
> > > | ED-Pose-Test (Swin-L)        | R.   |      30.3     |       37.7      |       33.5      |      86.7     |       98.6      |       93.0      |
> > > | ED-Pose-Finetune (Swin-L)    | R.   |      66.9     |  **74.3**  |  **71.0**  |      91.8     |       97.8      |       95.3      |
> > > | ED-Pose-Test (Swin-L*)       | R.   |      30.4     |       38.1      |       33.6      |      87.7     |  **98.9**  |       93.4      |
> > > | ED-Pose-Finetune (Swin-L*)   | R.   | **67.3** |       74.2      |  **71.0**  | **92.6** |       98.2      |  **96.0**  |

---

> > > > ### Comment · Reviewer_uhwo · 2022-11-21
> > > > **Response to Authors**
> > > >
> > > > Thank you to the authors for the rebuttal. Most of my concerns haven been addressed. I changed my recommendation to Accept.

---

### Official Review · Reviewer_vkvL · 2022-10-30

**Confidence:** 5
**Correctness:** 4
**Technical Novelty And Significance:** 2
**Empirical Novelty And Significance:** 3
**Recommendation:** 5

**Clarity, Quality, Novelty And Reproducibility:**

Paper is well written overall except some mistakes such as “cross-model” in page 2, etc. The approach is easy to reproduce. Novelty for the technical part is weak.

**Strength And Weaknesses:**

Strength:
1. A new large-scale Chinese cross-modal benchmark which makes great contribution to the community.
2. Different schemes are designed to learn better feature representations and overcome the problems of noisy labels.
3. The paper is well written and easy to read.
Weaknesses:
1. There is no substantive modification of the ranking algorithm. These two losses are not novel somehow compared with previous contrastive loss.
2. The distillation loss is not novel both on the target level and feature level. Some technologies such as " adopt soft targets generated by momentum-updated encoders" are common and there are no essential progresses over published approaches.


**Summary Of The Paper:**

  This paper proposes a large-scale Chinese cross-modal benchmark where one pre-training dataset and five fine-tuning datasets are involved. The new benchmark serves a significant contribution for the community. The paper proposes Global Contrastive Pre-Ranking and Fine-Grained Ranking to explore the matching relationship between image and text pairs. Besides, the paper employ distillation to overcome the problem of noisy labels.

**Summary Of The Review:**

Taking both the contributions from both technique and dataset, I recommend a borderline.

---

> ### Author Response · Authors · 2022-11-06
> **The Irrelevant Reviews to Our Paper**
>
> Dear Reviewer #vkvL,
>
> Thanks for your effort in reviewing the manuscript. Unfortunately, your comments are not relevant to our paper, and it is more like the comments to a paper about large-scale text-image pretraining. We are not sure if it is a system error or your misoperation. But anyway, could you please update your comments in the discussion phase? We would appreciate it and look forward to your reply.
>
> Best!
>
> The authors of Paper577

---

### Author Response · Authors · 2022-11-15
**General Response**

We thank the reviewers for their valuable and insightful feedback. We are encouraged that they found our paper well-written (\#uhwo, \#Dxnp, and \#EVzH), our approach novel (\#Dxnp and \#EVzH), and our experiments thorough (\#uhwo, \#Dxnp, and \#EVzH).

We have updated a revised version of our manuscript. The revised parts are highlighted in red and summarized below:

1. **Implementation details.** We provide more detailed loss functions and the corresponding loss coefficients (\#EVzH), the unique merit of ED-Pose for human detection pre-training (\#h7oS), and the detailed interactive learning between human detection and keypoint detection (\#uhwo) in Appendix-A.

2. **Inference time.** The original manuscript tested mainstream and recent methods (SimpleBaseline (TD), Poseur (TD), HrHRNet (BU), and PETR (OS)) on our A100 machine. To serve the follow-up work, we further test HRNet (TP), PRTR (TP), and InsPose (OS) on our A100 machine (\#ke Li) in Table. 3. Other methods still directly refer to PETR's paper due to no open-source code in Table. 8.

3. **More discussion.** We further discuss additional related works (\#uhwo) and analyze the advantage of ED-Pose (\#h7oS) in Appendix-D.

4. **Comprehensive code.** We provide more comprehensive code to reproduce our results under different backbones (i.e., ResNet-50 and Swin-L) and datasets (i.e., COCO and CrowdPose).

We appreciate all the suggestions made by reviewers to improve our work. We are looking forward to further feedbacks.

---

### Decision · Program_Chairs · 2023-01-20

**Decision:**

Accept: poster

**Justification For Why Not Higher Score:**

While the model seems to be interesting and quite effective, it is a variant of DETR for the multi-person pose task. Thus, AC recommends poster presentation.

**Justification For Why Not Lower Score:**

Though DETR part is an application, the human-to-keypoint decoder with the box-based keypoint formulation is a novel part.

**Metareview: Summary, Strengths And Weaknesses:**

The paper obtains 4 pieces of valid review reports, and the 5th one is invalid (comments are irrelevant to this paper). All the four reviewers are quite positive on this paper. The paper treats the task of multi-person pose estimation as two explicit box detection processes, which enables end-to-end training and achieves good pose estimation result. The method utilizes the DETR framework for pose estimation, and designed a human-to-keypoint decoder with the box-based key-point formulation, which seems to be interesting and effective. Authors have already revised some parts of the paper during rebuttal. Meanwhile, as pointed out by reviewers, more details need to be added to the paper and appendix. Thus, authors need to carefully incorporate the details and other comments into the camera ready version.

**Note From Pc:**

if the above contains the word "oral" or "spotlight" please see: "oral" presentation means -> notable-top-5% and "spotlight" means -> notable-top-25%. As stated in our emails, we are disassociating presentation type from AC recommendations